# Parametric Investigation on a Micro-Array Heat Sink with Staggered Trapezoidal Bumps

**DOI:** 10.3390/mi10120845

**Published:** 2019-12-04

**Authors:** Ruijin Wang, Weijia Yuan, Jiawei Wang, Zefei Zhu

**Affiliations:** School of Mechanical Engineering, Hangzhou Dianzi University, Hangzhou 310018, China; ywj24@hdu.edu.cn (W.Y.); wjw1129@hdu.edu.cn (J.W.)

**Keywords:** nanofluid, micro-array heat sink, staggered trapezoidal bumps, heat transfer enhancement

## Abstract

More efficient heat sinks are required due to the rapid increase of power density in microelectronic devices. In this study, a micro-array heat sink with stagger trapezoidal bumps was designed. Numerical simulations for the flow and heat transfer under various conditions were carried out to help us to fully understand the mechanisms of the heat transfer enhancement in such a heat sink. The effects of the structure of the heat sink, parameters of the bumps, and volume fraction of the nanofluid on the performance of heat sink were studied. The results show us that the bumps in the heat sink can result in chaotic convection, interrupting the thermal boundary layer and increasing the cooling area, subsequently improving the heat transfer performance. Furthermore, parametric investigations for trapezoidal bumps were conducted to obtain preferential values for parameters, such as the bump width, fore rake angle of the bump, bump height, and bump pitch.

## 1. Introduction

For the safety of microelectronic devices, a more efficient microchannel heat sink (MCHS) is necessary to reduce the temperature of devices because the miniaturization and integration of devices lead to a great increase in power density (frequently more than 100 W/cm^2^). MCHSs were proposed initially in the 1980s by Tuckeman [1] to take the place of the traditional heat exchanger for the cooling requirement of an integrated circuit (IC). In order to elevate the heat transfer performance, the structures and parameters of an MCHS, e.g., the shape of the microchannel [2,3], the width:depth ratio of the rectangular microchannel [4,5], the complex manifold geometries [6,7], the potential double/multilayers MCHS [8,9,10], the interrupted microchannel with various ribs [11,12], and so on, have been investigated. The most representative examples are as follows. Khodabandeh et al. [3] optimized the geometrical dimensions of an MCHS with a trapezoidal microchannel using constructal theory when a 1% water-Al_2_O_3_ nanofluid is used. For a constant pressure drop, a microchannel with a 70° side angle in the optimum state has the maximum total thermal conductivity. A heterogeneous two-phase model was employed by Wang et al. [13] to study the microchannel heat sink using an Al_2_O_3_-water nanofluid as a coolant in a photovoltaic system. The numerical results showed that the effects of flow field, volume fraction, and nanoparticle size on the heat transfer enhancement in an MCHS were governed by the ratio of Brownian diffusion and thermophoretic diffusion. In addition, the geometric parameters of an MCHS and the physical parameters of the nanofluid were optimized. Chai et al. [14] indicated that having offset ribs on sidewalls can result in a significant heat transfer enhancement. Zhong et al. [15] investigated the fluid flow and heat transfer in novel sinusoidal microchannels with alternating secondary branches inspired by oblique fins. The combination of Dean’s vortices induced by the wavy channel and the cross-channel mixing enabled by the secondary branches showed potential to further enhance the convective heat transfer. 

Another way to improve the heat transfer efficiency of an MCHS is to interrupt the thermal boundary using various lugs and/or ruts that are periodically located on the side wall or bottom wall [16,17,18,19,20]. Wang et al. [18] performed a parametric investigation on an MCHS with slanted rectangular ribs because they can induce chaotic convection. The parameters, such as attack angle, height, length, width, and pitch of the slant of the rectangular ribs were optimized by evaluating the average Nusselt number with an identical pump power. Li et al. [19] presented a novel MCHS with triangular cavities and rectangular ribs. The effects of the cavities and ribs on the Nusselt number and friction factor were investigated. Behnampour et al. [20] compared the heat transfer performance of MCHSs with trapezoidal, rectangular, and triangular shaped ribs by using a water-Ag nanofluid. The results indicated that among all the investigated rib forms, the triangular form had the best thermal performance evaluation criteria (PEC) values. 

Furthermore, a combination of various structures can be employed to promote the advantages and weaken the disadvantages of a single structure [21,22,23,24]. Ghani et al. [22] proposed a novel MCHS with sinusoidal cross-mark cavities and rectangular ribs. The results showed that the thermal performance of an a novel MCHS with sinusoidal cross-mark cavities and rectangular ribs (MC-SCRR) was the most superior owing to the combination of two important features: a significant reduction of the pressure drop and an increase of the flow turbulence. More interestingly, a new scheme for an MCHS with wavy porous fins was proposed by Lu et al. [25] for reducing the pressure drop and thermal resistance simultaneously. The remarkable reduction in the pressure drop came from the combination of the permeation and slip effects. The heat transfer enhancement was attributed to the combination fluid mixing using a Dean’s vortex and the forced permeation via a jet-like impingement. Sidik et al. [24] summarized that the passive structures that normally improved the heat transfer performance were: the channel curvature, surface roughness, flow disruption, out-of-plane mixing, secondary flow, re-entrant obstruction, and fluid additives. In general, the mechanisms used to enhance the heat transfer in an MCHS include: intensifying the chaotic convection, interrupting the thermal boundary layer, enlarging the cooling area, and improving the thermal conductivity of the working medium. 

A nanofluid with favorable thermal conductivity can be employed as a working medium. On the basis of a micromechanical analysis, Buongiorno [25] concluded that the most important slip mechanisms between the nanoparticles and fluids are Brownian motion and thermophoresis. In addition, a two-component and four-equation model was established. Minea et al. [26] numerically investigated the convective heat transfer characteristics using an Al_2_O_3_-water nanofluid and indicated that the heat transfer coefficient increased from 3.4% to 27.8%. Mohammed [27] investigated the heat transfer of a trapezoidal MCHS using different nanofluids and substrate materials and indicated that a glycerin-based nanofluid has a greater heat transfer coefficient for a substrate heat sink. Fani [28] considered the size effects of spherical nanoparticles on the thermal performance and pressure drop of a CuO-water nanofluid in a trapezoidal MCHS. The results signified that the heat transfer decreases with an increase in particle size, and the decrease is greater when the volume fraction is larger.

It is necessary to uncover the microscopic mechanisms of heat transfer enhancement in an MCHS to improve the thermal performance of MCHSs used in microdevices. Based on the mechanisms mentioned above, a micro-array heat sink (MAHS) with staggered trapezoidal bumps (STBs) was designed to intensify the chaotic convection, to interrupt the thermal boundary, and to enlarge the cooling area. An Al_2_O_3_-water nanofluid was employed as a working medium with a high thermal conductivity. In order to fully understand the mechanisms and improve the heat transfer performance, numerical simulations were carried out to be certain of the flow pattern in the MAHS and to evaluate the heat transfer performance using the Nusselt number at an identical pump power. 

## 2. Geometric Model 

Based on the above comprehensive analysis, an MAHS with stagger trapezoidal bumps arranged on both the top and bottom substrates was designed (Figure 1). To decrease the computational cost, a 4 × 4 trapezoidal bump array was arranged on both substrates. The length, width, and height of the MAHS were 6000 μm, 1200 μm, and 400 μm, respectively. Therefore, L1 × W = 4800 μm × 1200 μm represented the heating area, and the power density on the bottom substrate was 100 W/cm^2^. The thickness of the bottom substrate was set to be 100 μm by considering the solid–liquid conjugated heat transfer. Both the inlet and outlet were located on the upper substrate, and the diameters of the inlet and outlet were set to be 330 μm. 

The parameters of the trapezoidal bumps were: the length of base side a and top side c, the height hr, the fore rake angle α, the rear rake angle
β and the space between the two bumps in the x-direction (i.e., bump pitch) d. In the present work, α, hr/H, b/c, and d/a were parametrically investigated to obtain a set of preferential values, while β was fixed to be 75°, in line with the results in Tang et al. [9].

## 3. Numerical Analysis

### 3.1. Numerical Model

Considering the conjugated heat transfer of the solid–fluid, heat transfer in the fluid and solid should be involved. Aiming at the silicon substrate MAHS using a nanofluid as a working medium, several assumptions, such as no viscous dissipation, steady incompressible and laminar flow, and neglecting gravitation and radiation heat transfer should be made to simplify the numerical calculation. The governing equations for the fluid can be described as:Conservation of mass:(1)$$\nabla \cdot \vec{V}=0$$
where $\vec{V}$ is the velocity of the nanofluid.Conservation of momentum:(2)$$\rho_{nf}[\frac{\partial \vec{V}}{\partial t}+(\vec{V}\cdot \nabla )\vec{V}]=-\nabla p+\mu_{nf}\nabla^{2}\vec{V}$$
where $\rho_{nf}$ is the density of the nanofluid, p is the pressure, and $\mu_{nf}$ is the dynamic viscosity of the nanofluid. Conservation of energy:(3)$$\rho_{nf}c_{nf}\left[\frac{\partial T}{\partial t}+\vec{V}\cdot \nabla T\right]=\nabla \cdot k_{nf}\nabla T+\rho_{p}c_{p}[D_{B}\nabla_{\varphi }\cdot \nabla T+D_{T}\frac{\nabla T\cdot \nabla T}{T}]$$
where $c_{nf}$ is the specific heat, $k_{nf}$ is the thermal conductivity of the nanofluid, and T is the temperature. Furthermore,
(4)$$D_{B}=\frac{K_{B}T}{3\pi \mu_{f}d_{p}}$$
is the Brownian diffusion coefficient related to the nanoparticle diameter $d_{p}$, where $K_{B}=1.381\times 10^{-23}$. Also,
(5)$$D_{T}=0.26\frac{k_{f}}{2k_{f}+k_{p}}\frac{\mu_{f}}{\rho_{f}}\varphi$$
represents the thermophoretic diffusion coefficient related to the volume fraction of the nanoparticle φ, and $k_{f}$, $\rho_{f}$, and $\mu_{f}$ are thermal conductivity, density, and dynamic viscosity of the base liquid, respectively. $k_{p}$ is the thermal conductivity of the nanoparticle. Conservation of species:(6)$$\frac{\partial \varphi }{\partial t}+\vec{V}\cdot \nabla \varphi =\nabla \cdot [D_{B}\nabla_{\varphi }+D_{T}\frac{\nabla T}{T}].$$

Equations (1)–(6) exactly represent the famous two-component, four-equation model of nanofluids derived by Buongiorno [27].

For a solid, only the thermal conduction is taken into consideration:(7)$$k_{s}\nabla^{2}T=0$$
where $k_{s}$ is the thermal conductivity of the silicon substrate.

The thermal physical parameters for the nanofluid are related to that of the base liquid and Al_2_O_3_ nanoparticles (listed in Table 1), e.g., the viscosity, heat specific, density, and thermal conductivity of the nanofluid. They vary with the flow, the temperature, and the volume fraction of the nanoparticles. The dynamic viscosity of the base fluid is:(8)$$\mu_{f}=(2.414\times 10^{-5})\times 10^{\frac{247.8}{\left(T-140\right)}}$$

According to the Brinkman model, the dynamic viscosity of a nanofluid is [30]:(9)$$\mu_{nf}=(1+2.5\varphi )\mu_{f}.$$

The density and heat specific of a nanofluid can be written as [31]:(10)$$\rho_{nf}=\varphi \rho_{p}+(1-\varphi )\rho_{f}$$
(11)$${\left(c\rho \right)}_{nf}=\varphi {\left(c\rho \right)}_{p}+(1-\varphi ){\left(c\rho \right)}_{f}.$$

The thermal conductivity model for an Al_2_O_3_-water nanofluid derived by Chon [32] is given as:(12)$$\frac{K_{nf}}{K_{f}}=1+64.7\varphi^{0.75}(\frac{d_{f}}{d_{p}}{)}^{0.37}(\frac{k_{p}}{k_{f}}{)}^{0.75}Pr\;{Re}_{p}^{1.23}$$
where the diameter of water molecule $d_{f}=$ 0.28 nm, the Prandtl number $Pr=c_{f}\mu_{f}/k_{f}$, the particle Reynolds number $Re_{p}=\rho_{f}u_{B}d_{p}/\mu_{f}$, the Brownian velocity $u_{B}=k_{B}T/3\pi \mu_{f}d_{p}\lambda_{f}$, and the mean free path of the water molecule $\lambda_{f}$ is 0.17 nm [33].

### 3.2. Parameters for Evaluating the Thermal Performance

It is necessary to define relevant parameters for assessing the heat transfer performance of an MAHS. The Nusselt number (Nu) represents the heat transfer performance caused by convection and is given as:(13)$$Nu=\frac{hD}{k}$$
where D is hydraulic diameter and the heat transfer coefficient h can be written as:(14)$$h=\frac{qA_{q}}{A_{c}\left(T_{c}-T_{f}\right)}$$
where $A_{q}$ and $A_{c}$ are the heated area and conjugated area (i.e., solid–fluid interface area on bottom wall), respectively. The average temperature in the conjugated area and in the fluid in the channel are, respectively, given as: (15)$$T_{c}=\frac{\int TdA}{\int dA}\;\mathrm{and}\;T_{f}=\frac{\int T\rho_{f}dV}{\int \rho_{f}dV}$$

The friction coefficient denoting the resistance force is given as:(16)$$f=\frac{2\Delta pD}{L\rho_{f}v_{in}^{2}}$$
where, Δp is the pressure drop.

The pump power of an MAHS for steady flow can be defined as:(17)$$P_{p}=\Delta pv_{in}A_{in}$$
where the inlet area $A_{in}=\frac{\phi^{2}\pi }{4}$ and ϕ is the inlet diameter, which was set to be 330 μm in the present work. The average Nusselt number with an identical pump power was suggested to be a good option to evaluate the thermal performance of an MAHS [34] because the pump power, a very common index in practical engineering, represents the flow resistance, and the Nusselt number (or thermal resistance) represents heat transfer capability. In fact, the Nusselt number with an identical pump power can be used to replace the $Nu/f^{1/3}$ [12,18] to evaluate the thermal performance of the heat sink.

### 3.3. Validity of the Numerical Model

The purpose of the present work was to design an MAHS with a good thermal performance. Hence, numerical simulations were carried out to obtain preferential geometric parameters of an MAHS with staggered trapezoidal bumps.

In the simulation, no velocity-slip and no temperature-jump boundary conditions were imposed for all walls. The inlet velocity $v_{in}$ in the +x-direction was assumed to be uniform at the inlet, and “outflow” at the outlet. The initial temperature for the fluid and solid were both 300 K all over the micro-array. A constant heat flux was assumed. Therefore, a Neumann boundary condition $-k_{s}\frac{\partial t}{\partial y}=q$ was exerted on the bottom wall, where q = 100 W/cm^2^ on the bottom wall, and q = 0 on all other walls, the inlet, and the outlet. All surfaces of contact between the different materials were set to be an “interface”. 

Because of the interaction effects between the temperature, viscosity, flow, and volume fraction of the Al_2_O_3_ nanoparticles, the viscosity, density, specific heat capacity, thermal conductivity, and diffusion coefficient should be changed for every iteration. Hence, user define functions (UDFs) were programmed and compiled according to Equations (8)–(12). The SIMPLE algorithm was used to couple the velocity and pressure, and the second-order upwind scheme was applied in the numerical simulation.

The validity check needed to be carried out before the numerical model is to be employed in practical engineering. Numerical simulations for the calculation of heat transfer resistance was conducted, and the comparison of numerical results and published results in Ho et al. [35] show a good agreement (Figure 2), especially when the flow rate was larger than 400 cm^3^/min. 

## 4. Results and Discussions

First, the verification of grid independence was conducted. The discrepancy between the calculated thermal resistances for grid numbers of 805,123, 1,687,840, 2,503,300, and 4,352,610 was 2.63%, 1.01%, 0.03%, respectively, where the parameters of the trapezoidal bumps were: α=30°, hr/H=1, b/c=1, and d/a=0. Hence, the calculation accuracy was high enough when the grid number was bigger than 2,503,300. 

### 4.1. Effect of the Bump Structure

It was important for us to clarity the influence of bump structures on the heat transfer performance. Numerical simulations were carried out to compute the flow and heat transfer in an MAHS with STB when the Reynolds number ranged from 100 to 1000. The Nusselt number Nu and the friction coefficient f were calculated when water and a 2 vol% Al_2_O_3_-water nanofluid was used as the cooling medium (Figure 3). Note that a triangular bump occurs when c=0. It can be seen that the Nu increased with and increasing Reynolds number and the fiction coefficient decreased with an increasing Reynolds number. In addition, the Nu for a 2 vol% Al_2_O_3_-water nanofluid were a little higher than that for water (i.e., 0% nanofluid), while Nu for trapezoidal bumps was much greater than that for triangular bumps. Moreover, Nu for a heat sink with trapezoidal or triangular bumps was much greater than that without the bumps. In similar, the variation of friction coefficient shows us that, nanofluid is prior to water, trapezoidal bump is superior to triangular one. Hence, the influence of the flow pattern on the heat transfer in an MAHS was much greater than that of the cooling medium. However, thermal conduction improvement is one of the most well-known mechanisms for enhancing heat transfer, except for heat convection. Hence, a nanofluid should be investigated to determine the influence of additives in a coolant on the thermal performance of a heat sink. 

### 4.2. Effect of the Volume Fraction of the Nanoparticles

To understand the effect of the nanofluid on the heat transfer, the effects of the nanoparticle distribution, velocity, and temperature in an MAHS (Figure 4) were calculated. Figure 4a show us the nanoparticle concentration in the plane y = 350 μm. The lower concentration region near the base side of the bumps can be seen and was caused by the greater temperature gradient; subsequently, there was a greater thermophoretic force on the nanoparticles. The movement of nanoparticles away from the bottom enhanced the heat transfer due to the heat transportation by nanoparticles. Furthermore, the higher concentration region of nanoparticle existed at every corner of the bumpy bottom (Figure 4b) because the microvortex could trap the nanoparticles. Figure 4c shows that the global temperature rose along with the main stream due to the heat flux on the bottom substrate. The upward-moving fluids driven by the fore rake of the trapezoidal bumps on the bottom substrate was forced to mix with the downward-moving fluids driven by the fore rake of the trapezoidal bumps on the top substrate, which enhanced the heat transfer efficiency. Moreover, the nanoparticles near the bottom bumps enhanced the heat transfer because they were forced far away from the bottom bumps by the large temperature gradient. 

### 4.3. Overall Flow Patterns in the Micro-array

To reduce the computational cost, two 4 × 4 bump arrays arranged on the bottom and top substrates were employed to explore the mechanism of heat transfer enhancement in the MAHS. The streamlines the MAHS shown in Figure 5 were spiral lines. In detail, the fluids in the vicinity of the bottom wall (named “hot fluid”) were driven upward from the bottom to the center by the fore rake of the trapezoidal bumps on the bottom substrate. Meanwhile, the fluids in the vicinity of the top wall (named “cold fluid”) were driven downward from the top to the center by the fore rake of the trapezoidal bumps on the top substrate. Moreover, the fluids near the bottom wall were divided into two streams by the fore rake of the trapezoidal bumps on the substrates, one to the right and the other to the left. In the next half-cycle, what was just described retrograded. Subsequently, the streamlines became a spiral over the whole cycle. The spiral streamlines were beneficial for the creation of chaotic convection, and to elevate the heat transfer efficiency. It is worth noting that the arranged 4 × 4 bump array was suitable for simulation because the situation near the center row and column of the trapezoidal bumps was not affected by the boundaries (four sides).

### 4.4. Planar Flow Patterns

The flow patterns shown in Figure 4b cannot reveal the secondary flow that played an important role in the heat transfer. Hence, it was necessary to analyze the flow pattern in the plane normal to the main flow to understand the effect of the secondary flow on the heat transfer.

Figure 6 shows the flow patterns and temperature distributions with velocity vectors at various positions: x = 1300 μm (head of the fore rake of a bump, Figure 6a), x = 1400 μm (tail of the fore rake of a bump, Figure 6b), x = 1700 μm (head of the top edge of a trapezoid bump, Figure 6c), x = 1730 μm (tail of the top edge of a trapezoid bump, Figure 6d), x = 1760 μm (rear rake of a bump, Figure 6e), x = 1780 μm (rear rake of a bump, Figure 6f)). It is worth noting that only the central regions that were not affected by the boundary effect were investigated because the four sides were set to be adiabatic. Therefore, a minimum of 4 × 4 trapezoidal bump arrays were required in present work. Figure 6a,b shows that the fluid near the bottom was compelled by the fore rake of the bumps upward to the center region. In addition, the upward fluids divided into two streams, one to the left and the other to right. A pair-vortex was induced and grew gradually in the fore rake region. The pair-vortex grew to a maximum in the region of the top edge of the trapezoid (Figure 6c,d). The pair-vortex could transport the heat near the bottom to the center. Inversely, the bumps on the top substrate compelled the hot fluids from the center further to the top. A greater pair-vortex could enhance the heat transfer even more. The heat transfer efficiency of the trapezoidal bumps was greater than that of the triangular bumps because the pair-vortex in the MAHS with the trapezoidal bumps was larger than that with triangular ones. The pair-vortex faded away in the rear rake region and the heat transfer decreased. The pair-vortex was generated, grew, remained, and faded away in a cycle due to the bump array, with the continuous variations of the center of the pair-vortex having resulted in a blinking flow, strengthening the chaotic convection, and subsequently enhancing the heat transfer. Next, the analysis of the relationship between the temperature distribution and flow pattern was done. It can be seen from Figure 6c,d that the heat was brought from the bottom to the center by the pair-vortex induced by the bumps on the bottom substrate, and consequentially, from the center to the top by the pair-vortex induced by the bumps on the top substrate. Regrettably, almost no difference between the temperature distributions in Figure 6e,f existed near the bottom in the region of the rear rake of a bump, whereas there was an obvious difference between the temperature distributions in the center and in the top. As indicated, the heat diffused upward, following the flow. 

The distinction between the heat transfer performance of the trapezoidal bumps and the triangular bumps was analyzed in detail. The flow patterns and temperature distributions with velocity vectors at x = 1400 μm and x = 1700 μm are shown in Figure 7a,b. Like the flow pattern in the trapezoidal bump array, there was a pair-vortex in the triangular bump array as well. It is important to note that the pair-vortex in the triangular bump array was not large enough to drive the hot fluids near the bottom upward to a high region successfully. Therefore, the heat transfer performance of the trapezoidal bump array was better than that of the triangular bump array.

### 4.5. Nusselt Number along the Stream

The Nu along the stream, which represents the convection heat transfer, is shown in Figure 8. There were four waves corresponding to the four bump rows. The analysis of the variation of Nu (see the second wave in Figure 8) indicates that: the Nu went up steeply in the region of the fore rake of the trapezoidal bumps, remaining at a high value near the top side, and going down sharply near the rear rake. This can be interpreted as follows: more hot fluids near the bottom were driven upward to the center and more cold fluids were compelled downward to the bottom near the fore rake of the trapezoidal bumps. The Nu continuously increased because of the increasingly larger pair-vortex. The Nu did not increase any more near the top side of the trapezoidal bumps because the pair-vortex did not grow any more. The Nu went down slightly because the temperature of the fluid went up after the absorption of heat. The tendency was the reverse near the rear rake of the bumps because of the disappearance of the pair-vortex.

Of course, Figure 8 shows us the greater heat transfer enhancement in the trapezoidal bump array than in the triangular bump array. This agrees with the conclusion from the flow pattern analysis above. In addition, the waveform for the triangular bumps was somewhat different from that for the trapezoidal one. The Nusselt number increased at first and drops thereafter, with no plateau existing due to there being no top side for such a bump.

### 4.6. Parameterized Study of the Micro-Array 

It is known from the above analysis that the geometric parameters and Reynolds number greatly affect the heat transfer. The active influence of the trapezoidal bumps on the heat transfer was apparent, whereas the pressure drop greatly increased as well. An index that comprehensively considers the above two factors to assess the heat transfer performance of the MAHS is the global Nusselt number with an identical pump power.

#### 4.6.1. Effect of the Bump Width

Numerical simulations for various bump widths (*b*/*c* = 0.75, 1.0, 1.25, 1.5) were carried out to compute the Nusselt number with an identical pump power when *α* = 30°, 2*hr*/*H* = 1, and *d*/*a* = 0. Figure 9 shows the preferential value was about *b*/*c* = 1.25. The reasons were: (1) the cooling area doubled because the bumps on both the top and bottom substrates were connected together when *b*/*c* > 1.0; (2) the asymmetrical pair-vortex due to the asymmetrical geometry played a special role, i.e., the larger vortex in the pair-vortex could compel the smaller one to the center (Figure 10c); (3) the connected bumps (near the top side of a trapezoidal bump, Figure 10b,c) acted as the wall of the microchannel in the MCHS, and the fluids in other regions (fore and rear rake of the trapezoid, Figure 10a,d) were interconnected, as described in Chai et al. [11]; and (4) conversely, the greater the bump width, the greater the pressure drop, and subsequently, the greater the pump power. In a word, 1.0 < *b*/*c* < 1.25 was the range of preferential values.

#### 4.6.2. Effect of the Fore Rake Angle

In order to obtain the Nusselt number with an identical pump power when 2*hr*/*H* = 1, *d*/*a* = 0, and *b* = *c* = 200 μm, simulations for various fore rake angles α=15°,30°,45°,60° were carried out. Figure 11 indicates that the heat transfer performance reached a preferential value at α=45°. The reason was that the greater α could transport more hot fluids from the bottom upward to the center due to the bumps on the bottom and the intensification of the mixing of hot fluids and cold fluids from the top region driven by the top bumps. However, the flow resistance and pump power increased in the meantime. Thus, it can be concluded that a medium α = 45° was preferential.

#### 4.6.3. Effect of the Bump Height

Generally, higher bumps resulted in a greater cooling area and flow perturbance. Numerical simulations for various bump heights (2*hr*/*H* = 0.75, 1.0, 1.25, 1.5) were carried out when α = 45°, *d*/*a* = 0, *b* = *c* = 200 μm. The flow patterns for various bump heights are shown in Figure 12. It can be seen that the size of the pair-vortex increased when 2*hr*/*H* ≤ 1.0 and then decreased and disappeared gradually when 2*hr*/*H* ≥ 1.0. It seems that 2*hr*/*H* = 1.0 was the optimal value. However, the relation between the Nusselt number and the pump power shown in Figure 13 indicates that 2*hr*/*H* = 1.25 may be the most suitable value because the higher bump induced a greater pressure drop and a greater regression of the pair-vortex. It can be concluded after comprehensive consideration that 1.0 < 2*hr*/*H* < 1.25 may be preferential. 

#### 4.6.4. Effect of the Bump Pitch

The periodically arranged bumps produced spiral streamlines (shown in Figure 5) because the hot fluids near the bottom were driven upward by the bottom bumps in a certain row of the STBs, and were compelled downward by the top bumps in the next row back to the bottom. The pitch of the spiral streamlines was determined by the coordination of the bump pitch, fore rake angle, and inlet velocity. 

Smaller α, greater $v_{in}$ need to fit with greater d/a. Numerical simulations for various bump pitches (*d*/*a* = 0, 0.5, 1.0, 1.5) when *b* = *c* = 200 μm, *α* = 45°, 2*hr*/*H* = 1, L2 = 1800 μm. The results shown in Figure 14 indicate *d*/*a* = 0 is the preferential value. The reason is that, the reduction of bump length caused by the increase of bump pitch will downsize the region with pair-vortex, recede the chaotic mixing, and subsequently lower the Nusselt number.

## 5. Conclusions

Numerical investigations on the heat transfer of an MAHS with STBs were carried out when Al_2_O_3_-water nanofluid acted as a coolant. On this basis, the effects of the STBs on the internal flow and heat transfer enhancement were discussed. The preferential geometric parameters of trapezoidal bump were obtained. The following conclusions can be drawn: The underlying mechanisms to enhance the heat transfer of an MAHS with STBs were chaotic convection, interruption of the thermal boundary layer, and enlargement of the cooling area caused by the bump array.The heat transfer performance of an MAHS with a trapezoidal bump array was higher than that of an MAHS with a triangular bump array because the much larger pair-vortex near the top side of the trapezoidal bumps could drive the hot fluids upward, away from the bottom.The Nusselt number with an identical pump power, which considered both heat transfer and flow resistance, was suitable to evaluate the heat transfer performance. The numerical results indicated that the preferred values for the geometric parameters of the trapezoidal bumps were: 1.0 < *b*/*c* <1.25, *α* = 45°, 1.0 < 2*hr*/*H* < 1.25, and *d*/*a* = 0.

## Figures and Tables

**Figure 1 micromachines-10-00845-f001:**
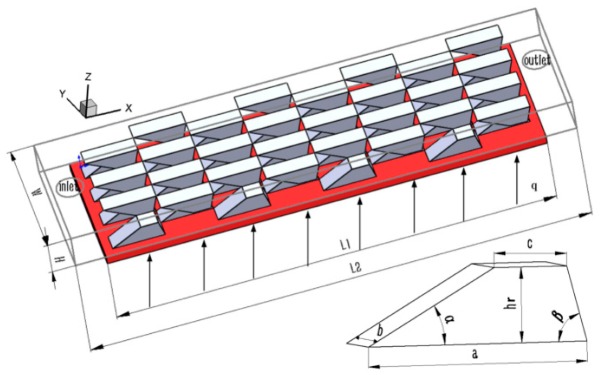
Structure and parameters of the stagger trapezoidal bump array.

**Figure 2 micromachines-10-00845-f002:**
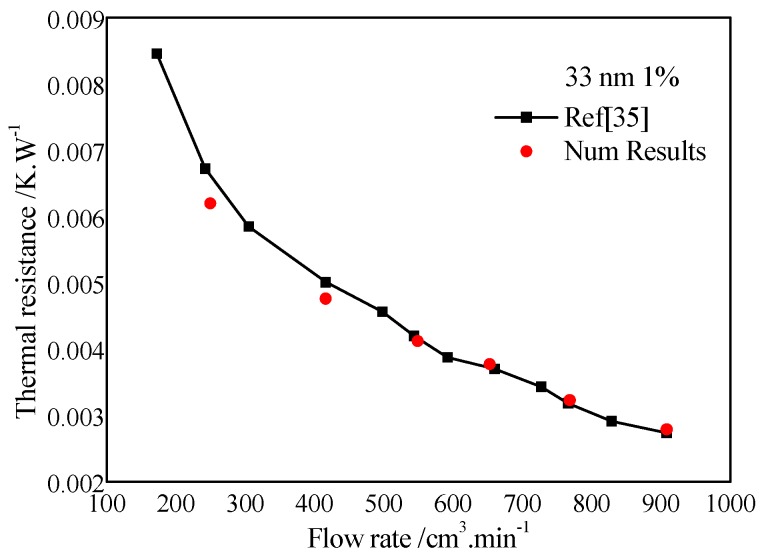
Validity of the numerical model.

**Figure 3 micromachines-10-00845-f003:**
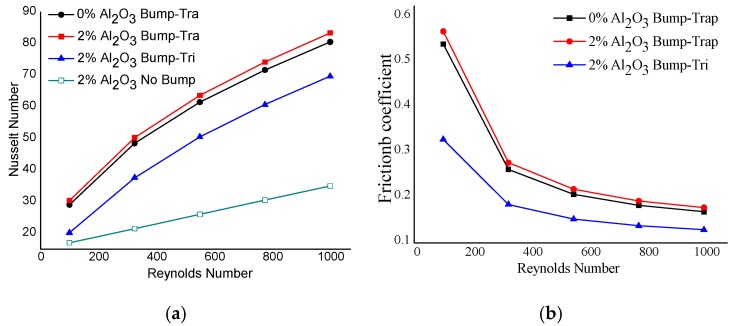
Nusselt number (**a**) and friction coefficient (**b**) vs. Re in MAHS with and without bumps array.

**Figure 4 micromachines-10-00845-f004:**
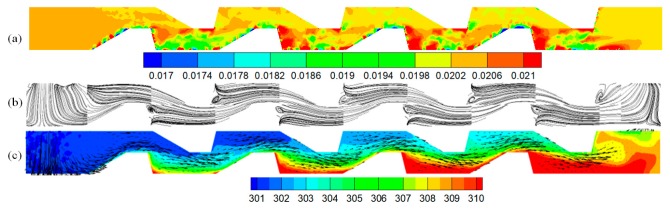
Nanoparticle distribution (**a**), flow pattern (**b**) and temperature contours with velocity vectors (**c**) in the plane y = 350 μm for 2 vol% nanofluid.

**Figure 5 micromachines-10-00845-f005:**
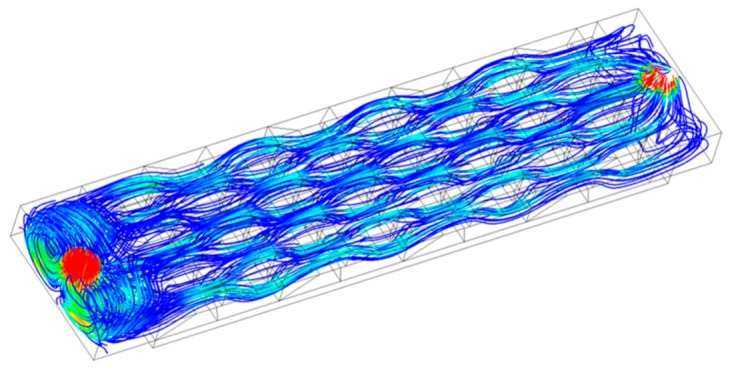
Streamlines in the whole region.

**Figure 6 micromachines-10-00845-f006:**
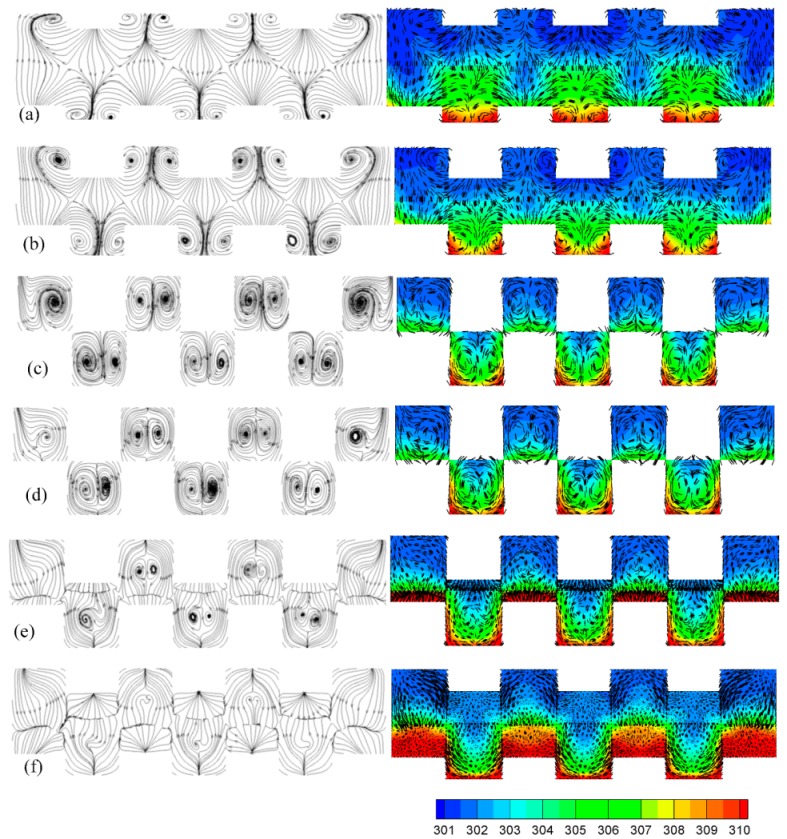
The flow patterns and temperature contours with velocity vectors at x = 1300 μm (**a**), x = 1400 μm (**b**), x = 1700 μm (**c**), x = 1730 μm (**d**), x = 1760 μm (**e**), x = 1780 μm (**f**) in trapezoidal rib array.

**Figure 7 micromachines-10-00845-f007:**
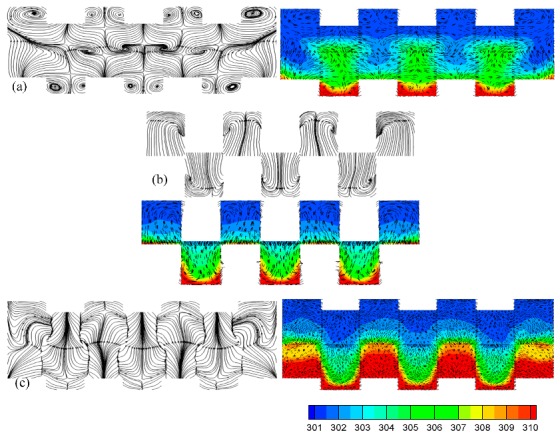
Flow patterns and temperature distributions with velocity vectors at x = 1400 μm (**a**), x = 1700 μm (**b**), x = 1780 μm (**c**) in triangular rib array.

**Figure 8 micromachines-10-00845-f008:**
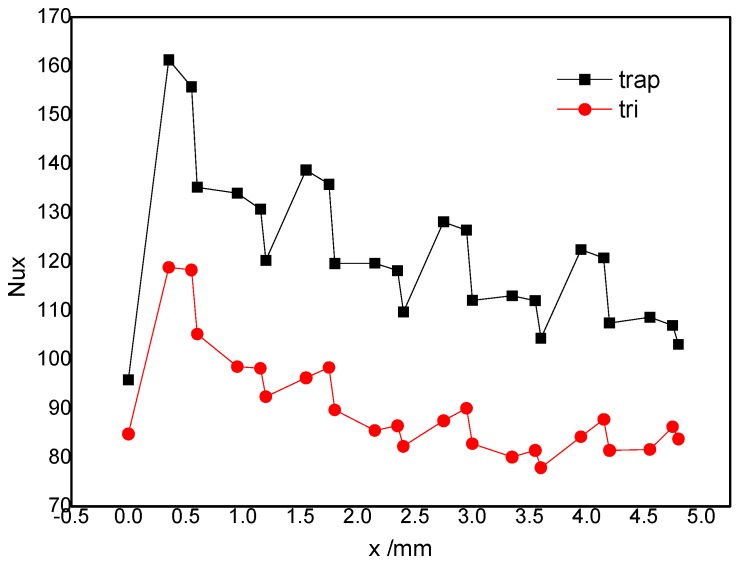
Nusselt number along the main flow.

**Figure 9 micromachines-10-00845-f009:**
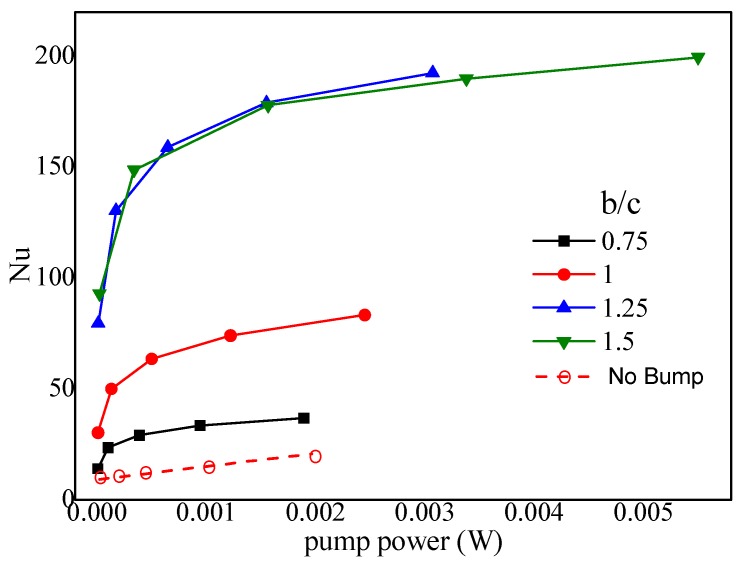
Effect of bump width on the Nusselts number vs. Pump power (*α* = 30°, 2*hr*/*H* = 1, *hr* = 0.2 mm, *d*/*a* = 0, *c* = 0.2 mm).

**Figure 10 micromachines-10-00845-f010:**
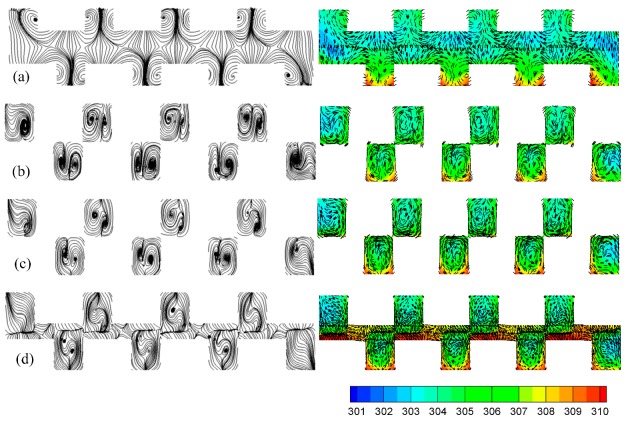
The flow patterns and temperature distributions with velocity vectors at various x in trapezoidal rib array x = 1400 μm (**a**), x = 1700 μm (**b**), x = 1730 μm (**c**), x = 1760 μm (**d**).

**Figure 11 micromachines-10-00845-f011:**
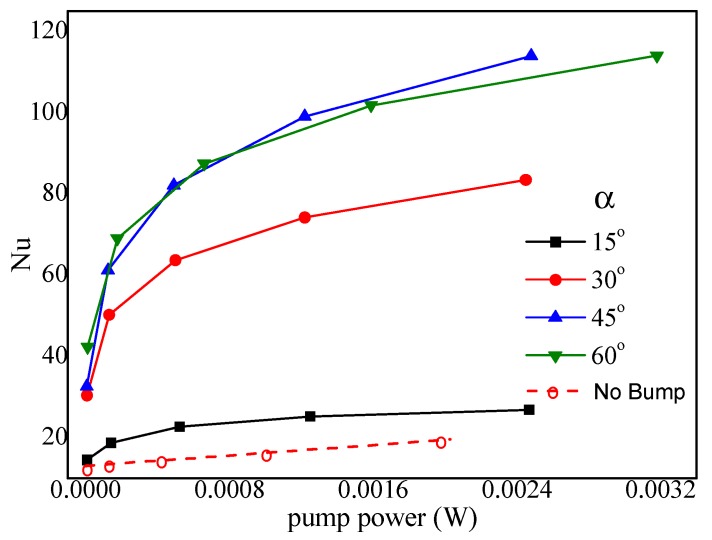
Effect of fore rake angle on the Nusselts number (*b*/*c* = 1, *c* = 0.2 mm, 2*hr*/*H* = 1, *d* = 0).

**Figure 12 micromachines-10-00845-f012:**
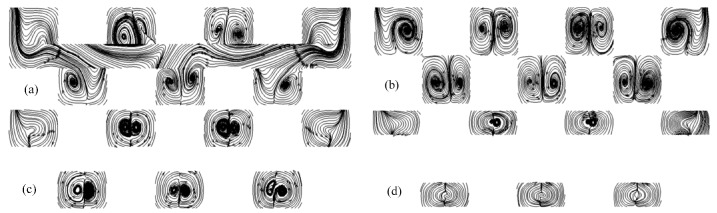
Effect of rib height on the flow pattern. 2*hr*/*H* = 0.75 (**a**), 2*hr*/*H* = 1 (**b**), 2*hr*/*H* = 1.25 (**c**), 2*hr*/*H* = 1.5 (**d**).

**Figure 13 micromachines-10-00845-f013:**
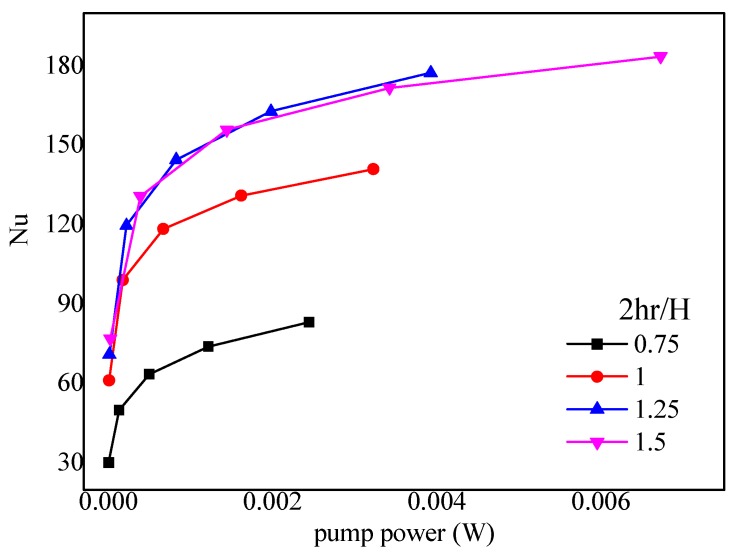
Effect of bump height on the Nusselts number vs. Pump power (*α* = 45°, *b*/*c* = 1, *hr* = 0.2 mm, *d*/*a* = 0).

**Figure 14 micromachines-10-00845-f014:**
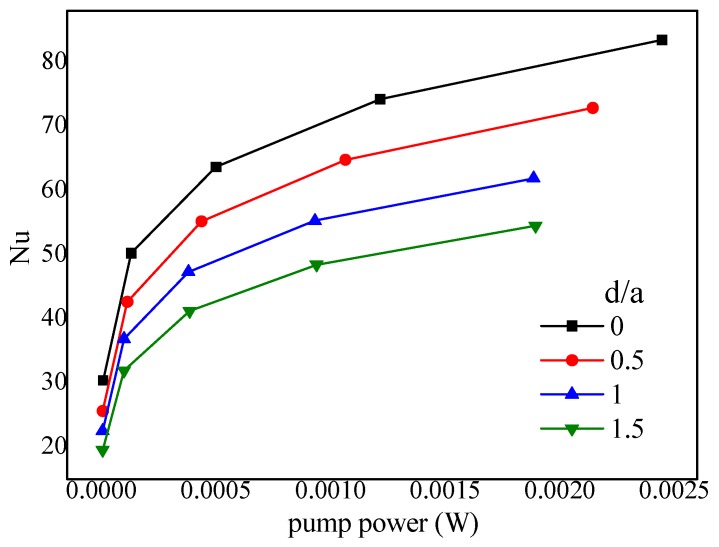
Effect of bump pitch on the Nusselts number vs. Pump power (*α* = 45°, 2*hr*/*H* = 1, *hr* = 0.2 mm, *b*/*c* = 1).

**Table 1 micromachines-10-00845-t001:** Thermal physical parameters for base liquid and nanoparticle [29].

Matter	Density *ρ* (kg/m^3^)	Thermal Conductivity k (W/m·K)	Specific Heat Capacity c_p_ (J/kg·K)
Water	998.2	0.6	4182
Al_2_O_3_	3970	42	880

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
