# Peer review of "Parametric Investigation on a Micro-Array Heat Sink with Staggered Trapezoidal Bumps"

_micromachines, 2019, doi:10.3390/mi10120845_

Round 1

Reviewer 1 Report

            The present manuscript shows a numerical investigation on the heat transfer of nano-fluid flows on a micro-array heat sink with stagger trapezoidal bumps. The authors occupied wide-range parametric sweeps on the design of this kind of heat sink and got a group of preferential geometric parameters. This work is original and helpful for the design of micro-array heat sinks. The following items need to be discussed:

(1)Figures 5, 7, 8 and 11 are not clear, especially for the velocity vectors.

(2)As shown in Fig 5, the nanoparticle distribution is not smooth. Is it because the mesh of the fluid region is not fine enough? Can you show the mesh or show a mesh-independence study?

(3)REF[37] cannot be found in the References list. This is an important reference for the validation of the current model.

(4)The coefficient of ‘heat transfer performance, Nu/f1/3’ is widely used in the analyses of enhanced heat transfer, which is the best dimensionless number to compare different heat sinks. However, it is not used in the present manuscript. The authors show the Nu vs. pump power in Figs 10 and 12, which is close to the sense of heat transfer performance. It is recommended that the authors use the more powerful heat transfer performance to analyze the effects of different parameters.

(5)A comparison between the current design and other traditional design is needed to show the superiority of the current design. The concept of ‘heat transfer performance, Nu/f1/3’ is recommended.

Author Response

Response to Reviewer 1

The present manuscript shows a numerical investigation on the heat transfer of nano-fluid flows on a micro-array heat sink with stagger trapezoidal bumps. The authors occupied wide-range parametric sweeps on the design of this kind of heat sink and got a group of preferential geometric parameters. This work is original and helpful for the design of micro-array heat sinks. The following items need to be discussed:

Figures 5, 7, 8 and 11 are not clear, especially for the velocity vectors.

Response: We redrew all of the velocity vector diagrams.

As shown in Fig 5, the nanoparticle distribution is not smooth. Is it because the mesh of the fluid region is not fine enough?Can you show the mesh or show a mesh-independence study?

Response: It is verified the mesh is fine enough in manuscript. However, we have collected only one value every 20 grid in figure process, so the nanoparticle distribution is not smooth enough. It is finer when we redrew it more accurately(see in revised manuscript).

REF[37] cannot be found in the References list. This is an important reference for the validation of the current model.

Response: I'm sorry, I have lost one cited literature. The lost reference is added.

The coefficient of ‘heat transfer performance, Nu/f1/3’ is widely used in the analyses of enhanced heat transfer, which is the best dimensionless number to compare different heat sinks. However, it is not used in the present manuscript. The authors show the Nu vs. pump power in Figs 10 and 12, which is close to the sense of heat transfer performance. It is recommended that the authors use the more powerful heat transfer performance to analyze the effects of different parameters.

Response: Yes, you are right. PEC criterion(Nu/f1/3) is a comprehensive index to evaluate heat transfer performance. However, Nusselt number at identical pump power is also a frequently-used index for heat transfer engineering[36] because it is equivalent to PEC. We discussed this issue in previous work[9]. Note, ,. Indeed, PEC is equal to Nu with identical pump power[12].

        A comparison between the current design and other traditional design is            needed to show the superiority of the current design. The concept of ‘heat          transfer performance, Nu/f1/3’ is recommended.

Response: Thanks for your suggestion. PEC is actually equivalent to Nu with identical pump power.

Reviewer 2 Report

In this study, a heat sink with stagger trapezoidal bump is designed and numerical simulations are performed to analyze the nanofluid flow and heat transfer under various conditions. Parametric study is done and the effects of geometry, and nanofluid’s volume fraction are investigated. In general, the paper explains the research clearly and is interesting. However, to make this manuscript suitable for publication the authors need to address the following comments:

The boundary conditions for the pressure field should be explained. Moreover, more information regarding the numerical simulation must be prepared. Is the SIMPLE algorithm used to couple the velocity and pressure? Is the 2nd order upwind scheme applied?

Equation 7, that shows dynamic viscosity of nanofluid, is very simple and is only applicable for very dilute nanofluids. Generally, for volume fraction higher than 1%, this correlation significantly underestimates the viscosity. We have more accurate correlations in the literature. The authors must explain why the above correlation is chosen.

Figure 4 shows that the results of pure water and 2% alumina-water nanofluid are almost the same. Here, the question is that why do we need nanofluid? In my opinion, we can simply use water and have the same cooling rate.

Author Response

Response to Reviewer 2

In this study, a heat sink with stagger trapezoidal bump is designed and numerical simulations are performed to analyze the nanofluid flow and heat transfer under various conditions. Parametric study is done and the effects of geometry, and nanofluid’s volume fraction are investigated. In general, the paper explains the research clearly and is interesting. However, to make this manuscript suitable for publication the authors need to address the following comments:

The boundary conditions for the pressure field should be explained. Moreover, more information regarding the numerical simulation must be prepared. Is the SIMPLE algorithm used to couple the velocity and pressure? Is the 2nd order upwind scheme applied?

Response: The SIMPLE algorithm is used to couple the velocity and pressure. The second order upwind scheme is applied. Some details are added.

Equation 7, that shows dynamic viscosity of nanofluid, is very simple and is only applicable for very dilute nanofluids. Generally, for volume fraction higher than 1%, this correlation significantly underestimates the viscosity. We have more accurate correlations in the literature. The authors must explain why the above correlation is chosen.

Response: The reason we chose this model in manuscript for viscosity of nanofluid is, Brinkman model is wide-used and simply model, and also be suggested for dilute nanofluid < 5vol% by many publications. Regretly, the comparison and selection of different models is really not considered in our work. We will select more accurate model for various nanofluid in future work. Thanks for your suggestion.

Figure 4 shows that the results of pure water and 2% alumina-water nanofluid are almost the same. Here, the question is that why do we need nanofluid? Inmy opinion, we can simply use water and have the same cooling rate.

Response: Yes, I agree. The reason we use nanofluid in heat sink is, the most well-known mechanisms to enhance heat transfer are to improve thermal convection and conduction. And additive in fluid(e.g. nanofluid) is an approach to elevate thermal conduction. We intend to understand the importance of the additives in coolant in present work.

In addition, it is known the heat transfer enhancement by nanofluid is related to many factors. In our simulations, the Brownian motion and thermophoretic diffusion are under consideration, and the thermal conductivity model of Chon is employed. This means all other mechanisms, such as aggregation, nanolayer and so on, aren’t taken into account. Hence, the heat transfer enhancement by nanofluid are very limited.

Furthermore, the dominated factors in present heat sink are chaotic convection and thermal boundary layer interruption. The main objective of present work is to design a microarray heat sink with higher thermal performance by arranged bump arrays.

Round 2

Reviewer 1 Report

(1) I required the authors to compare the performance of the current design with other traditional design in my last review. However, I cannot find this comparison in the R1 version of the manuscript. I insist that the authors should include this comparison as in the majority of this kind of paper. It makes sense even to compare with simple straight channels.

(2) I do not agree that ‘PEC is equal to Nu with identical pump power’ as the authors state in Line 147. The reasons are as follows:

They are not equivalent because one of them cannot be calculated from the other. In Ref[36], they normalized the pump power as f1/3Re, which makes much more sense than Figs 10 and 12 in the current manuscript. PEC is good to compare the performances of different designs. ‘Nu with identical pump power’ cannot do that unless the ‘pump power’ is normalized.

(3) The language needs editing.

Line 147: Indeed -> In fact, equal -> equivalent

Line 149: designed -> design

Line 155: Riemann boundary condition -> Neumann boundary condition

I just picked some mistakes randomly on Page 6. There are more through the manuscript. Please check the language more carefully.

Author Response

Response to Reviewer 1

Thanks for your comments.

(1) I required the authors to compare the performance of the current design with other traditional design in my last review. However, I cannot find this comparison in the R1 version of the manuscript. I insist that the authors should include this comparison as in the majority of this kind of paper. It makes sense even to compare with simple straight channels.

Response: Sorry. Why we didn’t add any comparison in R1 version is, the heat sink will be a rectangular flat cavity(only one channel) with one inlet and one outlet if there are no arranged array bumps. Nusselt number for MACH without bumps are added in R2 version.

(2) I do not agree that ‘PEC is equal to Nu with identical pump power’ as the authors state in Line 147. The reasons are as follows:

They are not equivalent because one of them cannot be calculated from the other. In Ref[36], they normalized the pump power as f1/3Re, which makes much more sense than Figs 10 and 12 in the current manuscript. PEC is good to compare the performances of different designs. ‘Nu with identical pump power’ cannot do that unless the ‘pump power’ is normalized.

Response: You are right. PEC is normally used to evaluate thermal performance for MCHS in most published works. Why did we(and many other researchers) select ‘Nu with identical pump power’ in present article? Because Nu (being inversely proportional to thermal resistance RT) presents heat transfer capability, pump power Pp presents flow resistance. Actually, ‘Nusselt number or thermal resistance with identical pump power’ are also very common index to evaluate thermal performance of MCHS.   

Furthermore, in Ref[36], they normalized the pump power as f1/3Re. It is known, f⇒Δp/u2,Pp⇒uΔp⇒fu3. So Pp1/3f1/3u⇒Ref1/3. In other words, f1/3Re is equivalent to (pump power). No doubt, whether Pp1/3or Pcan be used to evaluate the thermal performance of MCHS. In addition, pump power was a common index in engineering.

(3) The language needs editing.

Line 147: Indeed -> In fact, equal -> equivalent

Line 149: designed -> design

Line 155: Riemann boundary condition -> Neumann boundary condition

I just picked some mistakes randomly on Page 6. There are more through the manuscript. Please check the language more carefully.

Response:The manuscript is carefully checked and revised.

Reviewer 2 Report

No comments

Author Response

Thanks for your comments.

Round 3

Reviewer 1 Report

1. Pump power Pp 'presents' flow resistance qualitatively only rather than 'equivalent'. The author could state the reasons for using 'Nu with identical pump power'. But it is wrong to say 'Nusselt number with identical pump power is equivalent to PEC' as in Lines 149-150. They are not equivalent.

2. Please normalize the pump power in Figs. 10 and 12, otherwise there is no way to compare the current design to others.

3. The comparison between the current design and the straight channels should be shown in Figs. 10 and 12 to show the performance of the current design.

4. The language still needs editing. I can still find many grammatical mistakes in the present manuscript.

Author Response

Comments and Suggestions for Authors

1. Pump power Pp 'presents' flow resistance qualitatively only rather than 'equivalent'. The author could state the reasons for using 'Nu with identical pump power'. But it is wrong to say 'Nusselt number with identical pump power is equivalent to PEC' as in Lines 149-150. They are not equivalent.
 Response: Nusselt number with identical pump power can be used to replace the PEC to evaluate thermal performance of heat sink.
2. Please normalize the pump power in Figs. 10 and 12, otherwise there is no way to compare the current design to others.
Response: Comparisons can be carried out if we redraw Figs 10 and 12 by replacing Pp by Pp1/3.
3. The comparison between the current design and the straight channels should be shown in Figs. 10 and 12 to show the performance of the current design.
Response: Revised according to this comment.
4. The language still needs editing. I can still find many grammatical mistakes in the present manuscript.
Response: We checked and revised again.
